# Role of Metabolomics and Metagenomics in the Replacement of the High-Concentrate Diet with a High-Fiber Diet for Growing Yushan Pigs

**DOI:** 10.3390/ani14192893

**Published:** 2024-10-08

**Authors:** Min Xie, Dan Fei, Yelan Guang, Fuguang Xue, Jun Xu, Yaomin Zhou

**Affiliations:** 1Key Laboratory of Agro-Product Quality and Safety of Jiangxi Province, Institute for Quality & Safety and Standards of Agricultural Products Research, Jiangxi Academy of Agricultural Sciences, Nanchang 330200, China; zbsxiemin@126.com (M.X.); feidan@163.com (D.F.); guangyelan@163.com (Y.G.); 2Jiangxi Province Key Laboratory of Animal Nutrition/Engineering Research Center of Feed Development, Jiangxi Agricultural University, Nanchang 330045, China; xuefuguang1024@jxau.edu.cn; 3Animal Husbandry and Veterinary Science, Jiangxi Academy of Agricultural Sciences, Nanchang 330200, China

**Keywords:** energy, dietary fiber, growing swine, growth performance, fatty acids composition

## Abstract

**Simple Summary:**

The objective of this study was to investigate the regulatory effects of replacing a high-concentrate diet with a high-fiber content feed, which consists of wheat bran, rice bran, and corn bran, on the productive performance, meat quality, and fat acid composition. Our findings showed that increasing dietary fiber content significantly increased average daily feed intake (ADFI), up-regulated carbohydrate metabolism-related metabolites, proliferated abundances of fiber-degradable microbial communities, such as *Lactobacillus* and *Bifidobacterium*, and significantly down-regulated lipid metabolism and cofactors and vitamin metabolism. Our findings indicated that higher dietary fiber content significantly reduced dietary energy provision, effectively decreased the backfat and abdominal fat content of Yushan pigs through proliferating intestinal fiber-degradable bacteria, and up-regulating the hepatic lipolysis-related gene expression. This finding may provide an altered method for rearing domestic Chinese pigs.

**Abstract:**

The objective of this study was to investigate the regulatory effects of a high-fiber content feed on the productive performance, meat quality, and fat acid composition. A total of 18 120-day-old Yushan pigs with similar initial body weight were randomly allotted into high-concentrate diet (high energy, HE) and high-fiber diet (low energy, LE) treatments for the determination of regulatory effects on productive performance, meat quality, and fatty acid content. Further, blood metabolomic, gut microbiota, and liver energy-related gene expression measurements were used to investigate the underlying mechanisms. Results showed that the LE treatment significantly increased ADFI while decreasing carcass weight, fat percentage, and IMF. Metabolomic results showed that the high-fiber treatment significantly down-regulated metabolites that participated in lipid metabolism such as cyclic ADP-ribose and hippuric acid, while up-regulated metabolites were mainly enriched in nitrogen metabolism such as DL-arginine and propionylcarnitine (*p* < 0.05). Microbial results showed relative abundances of *Lactobacillus* and *Bifidobacterium* are significantly proliferated in the high-fiber feeding treatments (*p* < 0.05). Transcriptomic results showed that genes mainly enriched into the lipid metabolism are significantly up-regulated under the high-fiber dietary treatment (*p* < 0.05). Conclusion: higher dietary fiber significantly reduced dietary energy provision, effectively decreased the backfat and abdominal fat content of Yushan pigs through proliferating intestinal fiber-degradable bacteria, and up-regulating the hepatic lipolysis-related gene expression.

## 1. Introduction

Pork has long been considered the staple meat source for Chinese daily life [1]. Traditional Chinese black pigs generally exhibited higher backfat, intramuscular fat (IMF), and protein content, which provided a flavorful taste compared with the lean-type European and American pigs and gained the raising popularity in modern lifestyle [2,3,4,5]. Yushan black pig is a quintessential representation of traditional Chinese pigs, characterized by a big belly, higher protein (20.1–24.3%), intramuscular fat (2.01–3.61%), and unsaturated fatty acids (47.0–57.5%) contents [6]. However, the thickness of backfat and abdominal fat restricted feed efficiency and significantly increased feed consumption during the rearing process. Therefore, proper strategies that effectively reduce backfat and abdominal fat content significantly promote the utilization and popularity of Yushan pigs.

Dietary energy provision is the key factor that regulates the fat deposition process, and it varies efficiently due to the ingredient composition of feedstuffs. Previous studies showed a higher percentage of unsaturated fatty acids, such as C18:1 n-9, C18:2 n-6, C18:3 n-3, and total polyunsaturated fatty acids (PUFA), and a lower percentage of saturated fatty acids, including C16:0, C18:0, and C20:0, in the lower energy provision dietary treatment [7,8]. In addition, changes of feed nutritional components, particularly the dietary energy and fiber content, significantly impacted the nutrient digestibility and metabolic transformation and further impacted the fat deposition through modulating the physiological energy expenditure process [9,10].

Generally, brans and husks are representative sources of crude fiber, which effectively decline dietary energy content, remain resistant to digestive enzyme catalyzation, and reduce the absorption of fat and carbohydrates in animal GIT [11]. Feedstuff consumed by pigs that contains higher dietary fiber has been reported to effectively reduce the risk of diabetes, obesity, and gastrointestinal disorders [12]. Higher content of dietary fiber also functionally shaped the intestinal bacterial community and regulated the digestibility of other nutritional components. Alterations of bacterial communities in the pig gastro-intestinal tract (GIT) substantially influenced host physiological and immunological processes, which causatively regulated the growth and shaped the meat quality and flavor [13,14,15]. Additionally, an increase in dietary fiber content has been reported with a notable impact on hepatic lipo-metabolic-related gene expression and the fat deposition and composition of pigs [16]. Despite these findings, the underlying mechanism of increasing dietary fiber content on body composition was still of limited information. Further research is needed to fully understand the intricate relationship between dietary fiber, body composition, and overall health outcomes.

Therefore, this study aims to investigate the regulatory effects of lowering the energy provision induced by the higher content wheat bran, rice bran, and corn bran on the productive performance, meat quality, and fat acid composition. We hypothesized that higher dietary fiber content effectively decreased the backfat and abdominal fat content of Yushan pigs through enhancing intestinal fiber-degradable bacteria, regulating metabolic circulation, and up-regulating the hepatic lipolysis-related gene expression.

## 2. Materials and Methods

Animal care and procedures followed The Chinese Guidelines for Animal Welfare, which were approved by the Animal Care and Use Committee of the Jiangxi Academy of Agricultural Sciences; approval number 2024-JXAAS-XM-16.

### 2.1. Experiment Animals and Management

A total of 18 120-day-old Yushan pigs with similar initial body weight were selected and randomly allotted into low dietary fiber content (high energy, HE) and high dietary fiber content (low energy, LE) treatments for a 130-day-long feeding process, which contained a 10-day adaptive stage and a 120-day feeding stage. Each treatment contained 9 pigs, with each pig considered a replicate and reared in an individual pen that was 220 cm long × 90 cm wide × 130 cm heigh. Daily feed intake and daily weight gain were recorded individually. All pigs were reared in the same piggery, with all pens in a bidirectional arrangement. Feed was provided for each treatment twice per day at 07:30 and 18:30, and ingredients and chemical nutrients are displayed in Table 1. All pigs were allowed to get feed and water ad libitum. Productive procedures, which included immunization and cleaning, were conducted based on the regular rearing program. The relative humidity and temperature of the feeding piggery were maintained at 60–65% and 20–27 °C humidity and temperature, respectively.

### 2.2. Productive Performances and Carcass Performances Measurement

The feed intake (FI) of each pig was recorded and calculated by the deviation between the feeding scale and the residue before the next morning feed, and the average daily feed intake (ADFI) was calculated as the rate between total feed intake and the feeding days. The body weight of each pig was weighed at the end of the trial, and the average daily weight gain (ADG) was calculated through the following equation: ADG = (final body weight − initial body weight)/feeding days.

Next, the feed conversion ratio (FCR) between feed consumption and egg production was calculated based on the following equation:FCR = ADFI(g/d)/ADG(g/d).

At the end of the trial, all 18 pigs were slaughtered using CO_2_ stunning and bleeding in accordance with Chinese regulations for determining carcass performance of breeding pigs (NY/T 822-2004) [17] before a 12 h fast but with free access to water.

Carcass-related indexes, including carcass and length and backfat thickness, were measured. Carcass length was measured from the anterior edge of the first cervical vertebra to the posterior edge of the last lumbar vertebra. Backfat thickness was measured between the last cervical and first thoracic vertebrae and between the last thoracic and first lumbar vertebrae. The average of each measurement was calculated as backfat thickness. Organs including the heart, liver, spleen, and kidney were acquired and weighed; the organ indexes were calculated as the rate between organ weight and carcass weight. Furthermore, abdominal fat, subcutaneous fat, lean meat, skin, and bones were separated and weighed. The fat, lean, skin, and bone percentages were calculated using the following equation:(fat/lean/skin /bone) percentage = (fat/lean/skin /bone) weight(kg)/fat + lean + skin + bone weights(kg) × 100%

### 2.3. Meat Quality and Lipid Composition Measurement

Meat characteristics, which include the shear force, dripping loss, meat protein content (%), and intramuscular fat (IMF) (%), were measured based on the methods introduced in Chinese technical regulation for the determination of pork quality (NY/T 821-2019) [18]. Specifically, the meat was first cooked at a temperature of 72 °C and cut into rectangular, cooked meat sections (1 × 1 × 3 cm) when cooling to room temperature. Shear force was further measured perpendicular to the direction of fibers using a texture analyzer TA HD Plus (Stable Micro Systems Ltd., Surrey, UK) equipped with a Warner–Bratzler V-shaped shear blade (1.2 mm thick). Meat protein content was determined by the Dumas combustion method (method 992.15) [19]. IMF was determined through the ether extraction method, which was introduced by J. Folch [20].

The lipid composition was further analyzed according to the acetyl chloride methanol–methyl esterification method presented by S. Jaturasitha [21]. Both saturated fatty acid (SFA) and unsaturated fatty acid (UFA) were measured using the gas chromatograph Agilent 8860 GC, Santa Clara, CA, USA. Parameters of chromatographic measurement were set as follows: A dicyanopropyl polysiloxane column (100 m × 0.25 mm, 0.20 mm) was applied with a temperature of the column oven set at 140 °C for 5 min, and then gradually increased to 240 °C at 4 °C/min. The injector and detector temperatures were set as 260 °C and 280 °C, respectively. The chromatographic peak area was obtained by the integral method and further used for quantification as the content of lipid composition.

### 2.4. Blood Metabolomic Measurements

Five milliliters of each blood sample were collected during exsanguinations (using 10 mL EDTA tubes (Yuli medical instrument Co., Ltd., Nanjing, China)). Plasma was obtained by centrifugation at 2500× *g*, 4 °C for 10 min, and followingly applied for the blood metabolic measurement through the LC/MS analysis method.

To be simply stated, an internal standard was primarily made through the dissolution of 100 µL plasma by 400 µL 80% methanol and 0.02 mg/mL L-2-chlorophenyl alanine. All samples were following cleaned using ultrasound at 40 kHz for 30 min at 5 °C and carefully transferred to sample vials for LC-MS/MS analysis after the 13,000× *g* centrifugation at 4 °C for 15 min. Chromatographic identification of the metabolites was performed using a Thermo UHPLC system equipped with an ACQUITY UPLC HSS T3 (100 mm × 2.1 mm i.d., 1.8 µm; Waters, Milford, CT, USA). Mass spectrometric data were further collected using a Thermo UHPLC-Q Exactive HF-X mass spectrometer equipped with an electrospray ionization (ESI) source in either positive or negative ion mode. All raw data were imported into the Progenesis QI 2.3 (Nonlinear Dynamics, Waters, CT, USA) for peak detection and alignment and displayed into a data matrix that consisted of retention time (RT), mass-to-charge ratio (*m*/*z*) values, and peak intensity.

### 2.5. Gut Microbiota Analysis

Contents in the cecal section of each pig were sampled for 16S rRNA microbial diversity analysis. Briefly, cecal content DNA from each sample was extracted using the cetyltrimethylammonium bromide and sodium dodecyl sulfate (CTAB/SDS) method described by F. Xue [22], followed by the PCR amplification process using the primers designed through the V4 region of 520F (F: GTGCCAGCMGCCGCGGTAA) and 802R (R: GGACTACHVGGGTWTCTAAT). PCR amplification output of each sample was sequenced under the Illumina HiSeq 4000 platform (Illumina Inc., San Diego, CA, USA), and to control the raw tag quality a Quantitative Insights Into Microbial Ecology 2 (QIIME, V 2.0) package was used. Sequences within similarity > 97% were assigned to the same operational taxonomic unit (OTU). Alpha diversity, beta diversity, and functional prediction analyses were further conducted.

### 2.6. Liver Energy-Related Metabolic Gene Expression and Validation

Total RNA was first extracted from the liver samples of each pig using RNAiso Plus reagent (code No. 9109, Takara, Dalian, China). Purity and contamination of RNA were detected using 1% agarose gels and a NanoPhotometer^®^ spectrophotometer (IMPLEN, Westlake Village, CA, USA). RNA concentration was measured using a Qubit^®^ RNA Assay Kit with a Qubit^®^ 2.0 Fluorometer (Life Technologies, Carlsbad, CA, USA), followed by selecting 3 μg of RNA for transcriptome sequencing.

Then, the transcriptomic sequencing processes, which contained the amplification, library construction, sequencing, filtration, and gene identification, were conducted. PCR products were purified using the AMPure XP system, followed by the construction and quality assessment of the library using the Agilent 2100 Bioanalyzer system. The library was sequenced using the Illumina NovaSeq 6000 platform (Illumina Inc., San Diego, CA, USA). The quality of the RNA sequences was checked using FastQC (v0.11.9, Babraham Institute, www.bioinformatics.babraham.ac.uk (accessed on 21 May 2024)) and sequence adapters. Low-quality reads (read quality < 20) were removed, and the filtered sequenced reads were further mapped to the pig genome, followed by the quantification of the expression using the FeatureCounts (version: 1.5.0-p3) software.

The expression level of genes in each sample was first normalized and further conducted the differential analysis by the DESeq2 analysis package of R software (version 4.1.3, R Core Team, Vienna, Austria). Differential expressed genes (DEG) were filtered based on the criteria of |log2FoldChange|≥1 and *p* ≤ 0.05. KOBAS 3.0 (kobas.cbi.pku.edu.cn) was used to perform gene ontology (GO) and Kyoto Encyclopedia of Genes and Genomes (KEGG, http://www.genome.jp/kegg/, accessed on 22 May 2024) pathway enrichment analyses of the DEGs and association genes. A *p* value < 0.05 was considered significant in GO term enrichment and pathway analysis.

Total RNA was extracted using the RNA Simple Total RNA Kit (Tiangen, Tiangen Biochemical Technology Co., Ltd., Beijing, China) according to the manufacturer’s protocol. One microgram of total RNA was used to carry out reverse transcription using the Fast Quant RT Kit (Tiangen). Gene expression was quantitatively analyzed by qRT-PCR in a QuantStudio™ 5 Real-Time PCR System using AceQ qPCR SYBR Green Master Mix (Low ROX Premixed; Vazyme, Nanjing, China). The cycling conditions were as follows: 95 °C for 5 min, followed by 40 cycles of 95 °C for 10 s and 60 °C for 30 s. The relative mRNA expression of the target genes was calculated using the 2^−ΔΔCt^ method after normalization by the levels of GAPDH (a constitutively expressed gene that was used as the internal control).

### 2.7. Statistical Analysis

Productive performance, meat quality-related parameters, and intramuscular lipid content were firstly assessed for normal distribution using the SAS procedure “proc univariate data = test normal” and subsequently carried out the student’s *t*-test using SAS 9.4 (SAS Institute, Inc., Cary, NC, USA). Significance would be considered when *p* < 0.05, while a tendency towards significance was considered when 0.05 ≤ *p* < 0.10.

The normal distribution of relative OTU abundances of each sample was first assessed using the SAS procedure “proc univariate data = test normal”, followed by the differential analysis using the student’s *t*-test (SAS version 9.4, SAS Institute Inc., Cary, NC, USA). Alpha diversity and beta diversity of our samples were calculated with QIIME (version 1.7.0) and displayed with R software (version 4.1.3, R Core Team, Vienna, Austria). Principal coordinate analysis (PCoA) was displayed using the ggplot2 package in R software. Spearman correlations between bacteria communities and production performance were assessed using the PROC CORR procedure of SAS 9.4. Finally, the correlation matrix was created and visualized in a heatmap format using R software (version 4.1.3, R Core Team, Vienna, Austria).

Multivariate analyses on plasma metabolomic results, including principal component analysis (PCA) and orthogonal correction partial least squares discriminant analysis (OPLS-DA), were conducted using SIMCA-P software (V 14.0, Umetrics, Umea, Sweden). Differentially expressed metabolites between two treatments were identified based on variable importance in projection (VIP) from OPLS-DA analysis and statistical analysis (VIP > 1 and *p* < 0.05). Kyoto Encyclopedia of Genes and Genomes (KEGG, http://www.genome.jp/kegg/, accessed on 22 May 2024) was used to view the enriched pathways of different metabolites.

## 3. Results

### 3.1. Differential Analysis of Dietary Fiber Content on Growth Performance, Carcass Performance, and Meat Quality

The productive performance, including the ADFI, ADG, and the FCR, were calculated and are displayed in Table 2. A significant increment was discovered in ADFI (*p* < 0.05) in LE treatment compared with HE, while a decreased tendency of FCR was observed in the high-energy feeding treatment compared with the low-energy level feeding treatment (0.05 < *p* < 0.10). No other significant alterations were discovered for initial body weight, final body weight, or ADG.

All experimental pigs were chosen for the carcass performance measurement, and the results are shown in Table 3. The carcass weight and fat percentage were significantly higher, while the skin percentage was significantly lower in the HE treatment compared with LE (*p* < 0.05). No significant alterations were observed in leg weight, dressing percentage, lean percentage, and bone weight percentage (*p* > 0.05). Further, meat qualities such as shear force, dropping loss, meat protein content, and intramuscular fat (IMF) content were measured and results are shown in Table 3. The IMF content in HE treatment was significantly higher than that in LE treatment (*p* < 0.05). No significant changes were found for other parameters (*p* > 0.05).

### 3.2. Differential Analysis of High Fiber Content Dietary on Sarcous Fatty Acids Composition

The sarcous fatty acid composition, which includes both saturated and unsaturated fatty acids, was investigated, and the results are shown in Table 4.

C16:0 content was the only saturated fatty acid that significantly increased in HE treatment compared with LE (*p* < 0.05). In addition, the higher fiber content treatment significantly increased the content of C18:1(cis) while significantly decreasing the C18:1(trans) content compared with the higher concentrate diet treatment. No other significant differences are observed for the residual fatty acids between LE and HE treatments (*p* > 0.05).

### 3.3. Plasma Metabolic Responses to High Fiber Content Dietary Treatment of Yushan Pigs

Metabolic determination revealed a total of 623 across all samples through the filtering method, all these metabolites are listed in Appendix A. All identified metabolites were chosen for the differential analysis of integrative metabolic alteration between the HE and LE treatment through PCA and OPLS-DA analysis. As shown in Figure 1, PC1 and PC2 accounted for 31.5% and 14.7% of the total variation, respectively. Metabolites identified in the HE and LE treatment samples clustered into two clearly separated sections in both PCA and OPLS-DA analysis results, which means a significant difference between HE and LE treatments regarding plasma metabolites.

Furthermore, specifically differentially expressed metabolites between HE and LE treatments were identified based on the statistical standard of fold change > 2, VIP > 1, and *p* < 0.05. As shown in Table 5, a total of 10 down-regulated and 17 up-regulated metabolites were identified in HE treatment compared with LE. The up-regulated metabolites mainly contained cyclic ADP-ribose, 7-methylguanine, S-adenosylhomocysteine, adenosine, triiodothyronine, and hippuric acid, while the down-regulated metabolites mainly consist of DL-Arginine, ciprostene, propionylcarnitine, and desloratadine. No other significantly altered metabolites were found between the HE and LE treatments.

Functional enrichment and pathway analysis were applied based on the differentially identified compounds. All results are shown in Figure 2 and Figure 3. As Figure 2 shows, differentially expressed metabolites are functionally enriched in the carbohydrate metabolism, energy metabolism, amino acid metabolism, cofactors, and nucleotide metabolism. Specifically, energy metabolism showed a significant decline in higher fiber dietary treatment compared with HE treatment (*p* < 0.05). KEGG results showed that glycerophospholipid metabolism, nicotinate and nicotinamide metabolism, and bile secretion are the most enriched three pathways based on the differential metabolic content. Specifically, the pathway of regulation of the actin cytoskeleton is more enriched between LE and HE differential metabolites; however, the enriched number was low.

### 3.4. Effects of High Fiber Content Dietary Treatment on Gastrointestinal Bacteria Communities

Cecal bacterial communities were detected for the causative investigation on the productive and metabolic differences between HE and LE treatments. A total of 17 phyla and more than 250 genera were identified, and all information is shown in Appendix A. All identified microbial communities were selected for further analysis.

#### 3.4.1. α-diversity

Alpha diversity was first applied to detect the modulative effects of energy on bacterial abundances, and results are shown in Table 6. Bacterial diversities showed no significant alterations between HE and LE treatments through all parameters, including the Sobs, Shannon, Simpson, Chao, ACE, Pielou, and PD indexes (*p* > 0.05).

#### 3.4.2. β-diversity

Differential analyses on cecal bacteria between HE and LE treatments were followingly applied, and the result is displayed through PCoA. As shown in Figure 4, PCoA axes 1 and 2 accounted for 71.19% and 14.69% of the total variation, respectively. Bacteria communities between HE and LE treatments could be clearly separated by PCo1 and PCo2, except HE-5.

Differential analysis of the relative abundances of cecal bacteria at phyla and genera levels was calculated, and the results are shown in Table 7 and Table 8. At the phyla level, *Firmicutes*, *Bacteroidota*, and *Spirochaetota* accounted for the top three abundant phyla. Relative abundances of *Bacteroidota* and *Spirochaetota* significantly decreased, while *Firmicutes* significantly increased in HE treatment compared with LE treatment (*p* < 0.05). No other significantly altered phylum was observed between HE and LE.

At the level of genera, *Bacteroides*, *Lactobacillus*, *Rummeliibacillus*, *Prevotellaceae*, and *Faecalibacterium* contributed to the top 5 most abundant genera. Abundances of *Bacteroides*, *Prevotellaceae_UCG*, and *Clostridium* significantly increased in HE treatment (*p* < 0.05). Meanwhile, abundances of *Lactobacillus*, *Ruminococcus*, *Romboutsia*, *Succinivibrio*, *Phascolarctobacterium*, and *Bifidobacterium* are significantly proliferated in LE feeding treatment (*p* < 0.05). No significant changes were detected among other genera (*p* > 0.05).

### 3.5. Functional Prediction on the Differential Gut Microbiota

Predictive functions of the above-mentioned differential microbiota between HE and LE treatments were conducted through the Tax4Fun process [23], and the result is shown in Figure 5.

The predictive functional results include the metabolism, genetic information processing, environmental information processing, cellular processes, and organismal systems. Metabolism processes are the most impacted pathways, which include the carbohydrate metabolism, energy metabolism, amino acid, cofactors, and lipid metabolism. In addition, translation in genetic information processing and membrane transport in environmental information processing are also the most impacted functional processes enriched by the differential bacterial communities between high-fiber dietary treatment and high-concentrate diet treatments.

Moreover, the interactive effects between significantly altered gut bacteria and the significantly altered metabolites were selected for a correlation analysis, and all results are shown in Figure 6.

Microbial communities gathered into two main clusters. One was mainly composed of *Bacteroides*, *Rummeliibacillus*, *Enterococcus*, *Escherichia-Shigella*, and *Clostridium*, which showed a significantly positive correlation with ciprostene and 7-aminoflunitrazepam-d7 and a significantly negative correlation with 7-methylguanine, S-adenosylhomocysteine, adenosine, and hippuric acid. The other cluster mainly consisted of *Lactobacillus*, *Faecalibacterium*, *Ruminococcus*, *Bifidobacterium*, and *Fibrobacter*, which showed a completely inverse correlation compared with the former, as evidenced by the significantly positive correlation with hippuric acid, carbamazepine-d10, diflucortolone pivalate, and diacerein, as well as the significantly negative correlation with DL-arginine, desloratadine, and propionylcarnitine. Specifically, *Bifidobacterium*, which was considered the probiotic bacteria, showed a significantly positive correlation with all up-regulated metabolites in LE treatment compared with HE. No other significant correlations were observed.

### 3.6. Liver Energy-Related Metabolic Gene Expression Responses to High Fiber Content Dietary Treatment

Differential analyses of the liver energy-related metabolic genes were carried out. A total of 293 significantly differentially expressed genes based on the filter standard of fold change > 2, and *p* < 0.05 were identified, including 165 significant down-regulated and 129 significant up-regulated genes between HE and LE treatments. All these genes are list in Appendix A.

In addition, four energy metabolism-related genes, including two up-regulated and two down-regulated genes, were selected to verify the validity of the transcriptomic results (Figure 7). As the results show, the up-regulated genes of the protein phosphatase, Mg^2+^/Mn^2+^ dependent 1 K (*PPM1K*), and the protein phosphatase 1 inhibitor 3C (*PPR3C*) in the transcriptomic results of the HE treatment were significantly higher compared with the LE treatment. Similarly, the down-regulated genes of GTPase activating Rap/RanGAP domain like 3 (*GARNL3*) and growth arrest and DNA damage-inducible protein GADD45 beta (*GADD45B*) also showed the same expressing alteration (*p* < 0.05).

Finally, all differentially expressed genes were selected for the functional analysis, and the results are shown in Figure 8. As results show, the up-regulated genes in high-fiber dietary-treated livers are mainly enriched in metabolism pathways, which include the fatty acid metabolism, PPAR signaling pathway, glutathione metabolism, and fatty acid degradation. Additionally, the down-regulated genes of low-energy-treated livers are mainly enriched into the circadian rhythm, glycolysis, and apelin signaling pathways. Specifically speaking, the highest number of genes (140) enriched into the metabolic pathway; however, not significant. No other significant pathways are enriched by the significantly differentially expressed genes.

## 4. Discussion

### 4.1. Modulatory Effects of Energy Supplementation on Growth Performance

Growth performance has directly influenced the health and nutrient absorption in the pig production industry. In the present study, higher bran supplements significantly lower energy provision while increasing the ADFI compared with HE treatment. This finding is in line with X. Han [24], and the causal elements that modulated feed usage might be attributed to the following aspects.

Lower digestibility of dietary fiber digestibility (<50%) in pig gastrointestinal tract than other nutrients such as starch, sugars, fat, or protein (digestibility ranged from 80–100% in the pig gastrointestinal tract) may be the primary cause that increased ADFI [25,26]. The increment of dietary fiber content may indicate a higher feed intake to fulfill the nutritional requirement and therefore lead to an increase in ADFI. Additionally, the altered bacterial communities may further increase ADFI. In our study, relative abundances of fiber-degrading bacteria *Lactobacillus*, *Phascolarctobacterium*, and *Bifidobacterium* proliferated significantly, while the mainly energy provision bacterial community *Firmicutes* were significantly lower in LE feeding treatment [27]. These alterations help improve the fiber degradability and therefore increase ADFI in the LE treatment.

Further, the LE treatment pigs may have a higher anti-stressing capacity and immunity due to the higher expression of GADD45B, which functionally regulated the body growth and cell apoptosis by inhibiting the c-Jun N-terminal kinase (JNK) activation, suppressing IL-1-induced apoptosis, and activating the MAPK signaling pathway [28]. In the present study, higher expression of GADD45B was observed; this finding may further improve the anti-stressing capacity and help maintain the structural integrity of intestinal epithelial cells, which indirectly promoted the ADFI in the LE treatment. Probiotics, including *Lactobacillus* and *Bifidobacterium*, which played important roles in maintaining intestinal homeostasis and fortifying the intestinal mucus layer, significantly increased with the increment of dietary fiber content. A higher abundance of cecal *Lactobacillus* positively induced better feed efficiency, while *Bifidobacterium* help reduce pernicious bacteria proliferation [29]. Moreover, the increased abundance of probiotics in LE treatment promoted the transportation of epithelial cells, which further positively induced nutrient utilization and helped promote the ADFI in LE treatment.

### 4.2. Modulatory Effects of Energy Supplementation on Meat Quality

A higher lean meat percentage was acquired in the higher fiber content treatment. As previous studies show, lean meat was mainly formed through protein utilization. Physiological protein synthesis was mainly derived from the dietary crude protein degradation and amino acid utilization. In the gastrointestinal tract, a higher bacterial cellulose-degrading process was proved to increase meat protein content [30]. The increased fiber content in LE treatment significantly increased cellulose-degrading bacteria and provided more nutrients and energy for the protein synthesis process. This may partially be the cause of the increased lean meat percentage.

Carcass weight, fat percentage, and IMF content significantly decreased in the LE treatment, which was in line with X. Cui [31], whose findings revealed the decreased fat, moisture content, and weight loss in higher fiber content dietary treatments. Generally, the chemical composition of meat was greatly influenced by the type of fiber source used and its proportionate level in the product. The higher provision of dietary fiber in the LE treatment declined carbohydrate metabolic and energy provision rates, and causatively induced the decrease of fat percentage and IMF content.

### 4.3. Fatty Acid Composition and Regulatory Mechanism

Fat deposition was remarkably impacted by the dietary composition and energy supply. In our study, fat percentage, IMF, and saturated fatty acids such as C16:0 content in pork significantly decreased, while unsaturated fatty acid content increased in LE treatment. Our findings were similar to those of H. Wang [32], who observed the saturated fatty acid (SFA) content decreased and the total unsaturated fatty acid (USFA) content increased significantly with an increase in the level of rye bran. These alterations in fatty acid composition might be attributed to the following aspects.

First, the synthesis of fatty acids is an energy-consuming process that relies on the availability of ATP to catalyze the synthetic process and requires short-chain fatty acids (SCFAs) as the substrates [33]. SCFAs are primarily generated from carbohydrate degradation; the increasing of slower degradable dietary fiber content compared with starch supplied fewer SCFAs for fat synthesis [34], which further induced the fat deposition decline.

Additionally, interactions between gut microbiota and physiological metabolites might also down-regulate lipid deposition. *Butyricimonas* was reported with the higher activity of C18:0-forming bacteria [35], which decreased in LE treatment. This alteration might contribute to the decrease in saturated fatty acid content. In addition, increased relative abundances of *Lactobacillus* and *Bifidobacterium* are also reported to significantly reduce fat deposition and mainly work through the IGF-1 pathway [36,37]. The significantly increased *Lactobacillus* and *Bifidobacterium* further decreased the fat deposition of Yushan pig.

Moreover, the lower energy supply treatment further impacted the hepatic lipid metabolism, declined the lipid content, and altered the composition. The liver is primarily responsible for the physiological lipid metabolism and maintains the homeostasis between adipogenesis and lipolysis by regulating the hepatic gene expression. In our study, hepatic lipid utilization and the energy provision-related gene GARNL3 significantly up-regulated after the higher fiber-content dietary treatment, which further induced the improvement of lipolysis [38] and thus reduced the fat percentage and saturated fatty acids.

Despite the fact that an increased fiber content effectively modulates the meat quality of Yushan pigs by regulating the energy metabolism and lipid synthesis, further research is needed to determine the optimal dietary energy level allowing Yushan pigs to gain a higher meat quality, lean percentage, and meat production. In addition, while several microbial communities at the genera level have been identified, more accurate and in-depth analyses of the functions of differentially proliferated bacterial communities using the metagenomic and metatranscriptomic methods are required.

## 5. Conclusions

In summary, higher dietary fiber content significantly reduced dietary energy provision, effectively decreased the backfat and abdominal fat content of Yushan pigs by increasing intestinal fiber-degradable bacteria, and up-regulating the hepatic lipolysis-related gene expression.

## Figures and Tables

**Figure 1 animals-14-02893-f001:**
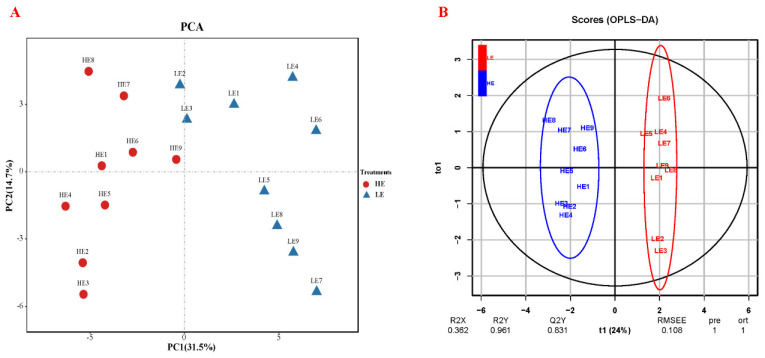
Integrative differential analysis of the plasma metabolic contents between HE and LE treatments. (**A**) Principal component analysis of plasma metabolites between the HE and LE treatment Yushan pig. (**B**) Orthogonal partial least squares-discriminant analysis (OPLS-DA) on plasma metabolites between the HE and LE treatment Yushan pig. HE = low-fiber content (high energy) treatment, LE = high-fiber content (low energy) treatment.

**Figure 2 animals-14-02893-f002:**
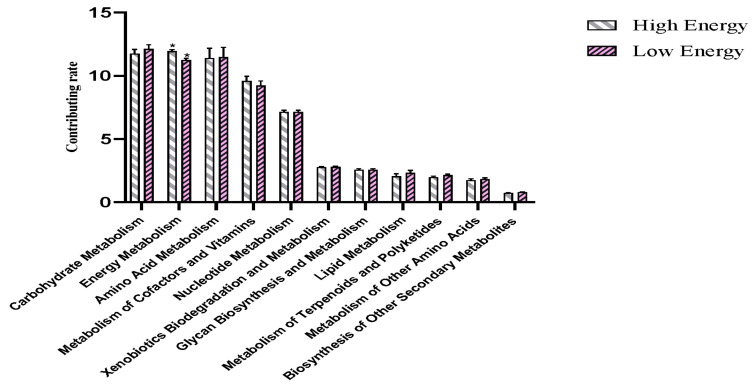
Functional enrichment analysis of differential metabolites between the high- and low-energy treatment Yushan pigs.

**Figure 3 animals-14-02893-f003:**
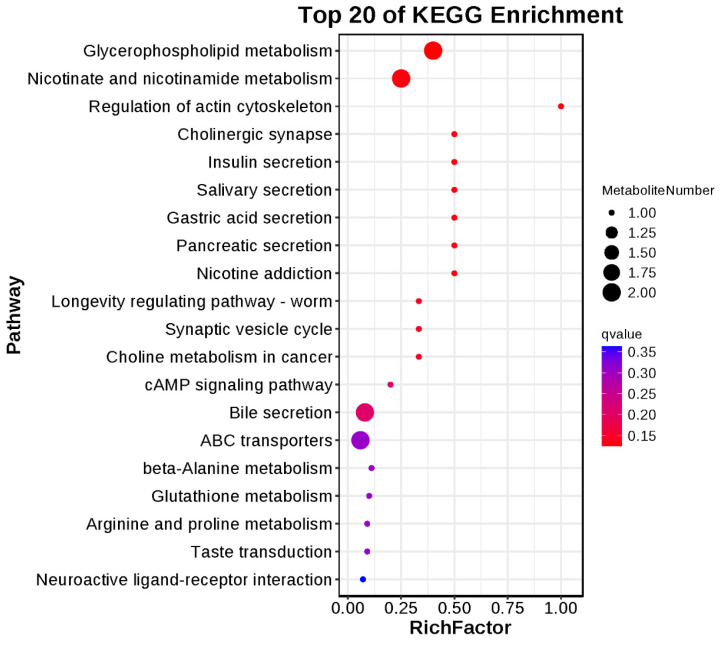
KEGG pathway analysis of differential metabolites between the high- and low-energy treatment Yushan pigs.

**Figure 4 animals-14-02893-f004:**
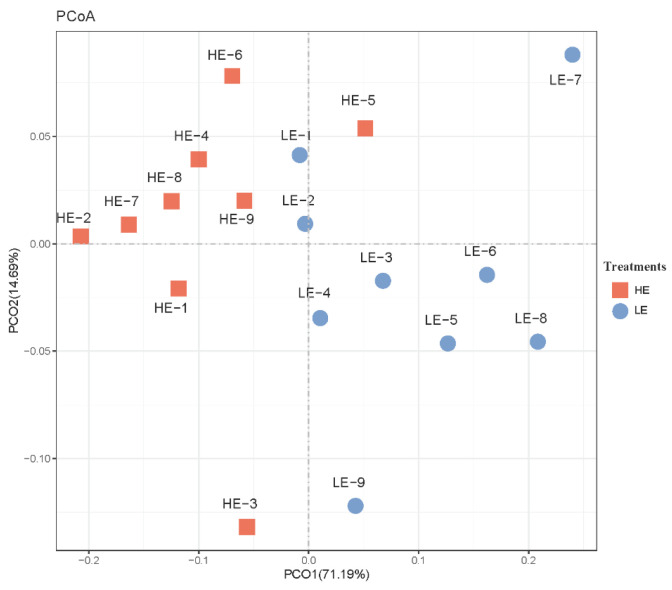
Principal coordinate analysis (PCoA) on community structures of the cecal microbiota between high- and low-energy treatment Yushan pigs. HE = low-fiber content (high energy) treatment, LE = high-fiber content (low energy) treatment.

**Figure 5 animals-14-02893-f005:**
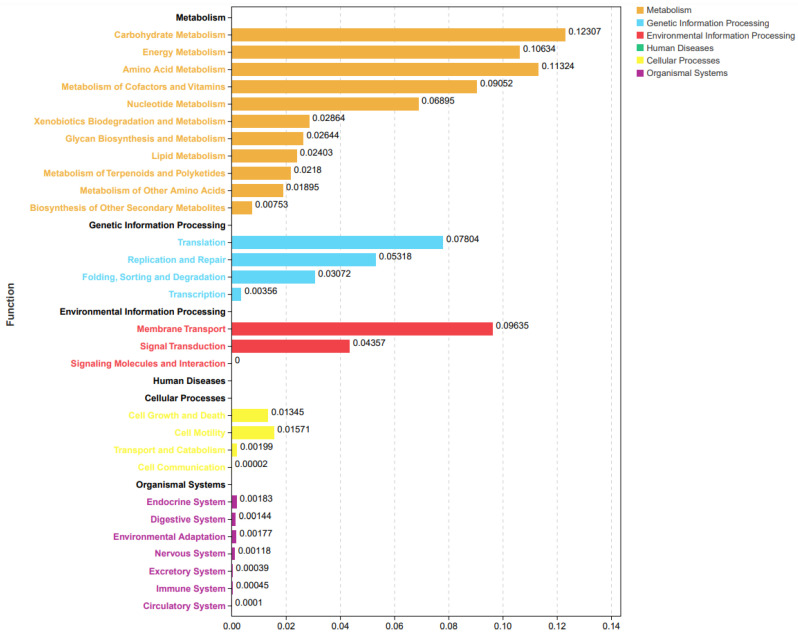
Tax4Fun functional prediction analysis of the differential abundant bacterial communities between high- and low-energy treatment Yushan pigs. HE = low-fiber content (high energy) treatment, LE = high-fiber content (low energy) treatment.

**Figure 6 animals-14-02893-f006:**
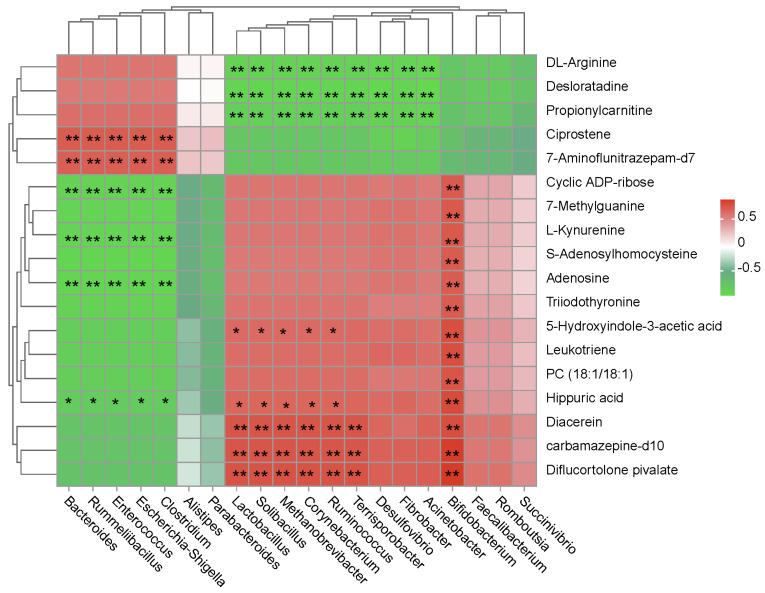
Correlation analysis between the significantly differential gut bacteria and blood metabolites. The red color represents a positive correlation, while the green color represents a negative correlation. “*” means a significant correlation (|r| > 0.55, *p* < 0.05). “**” means an extraordinary significant correlation (|r| > 0.75, *p* < 0.01).

**Figure 7 animals-14-02893-f007:**
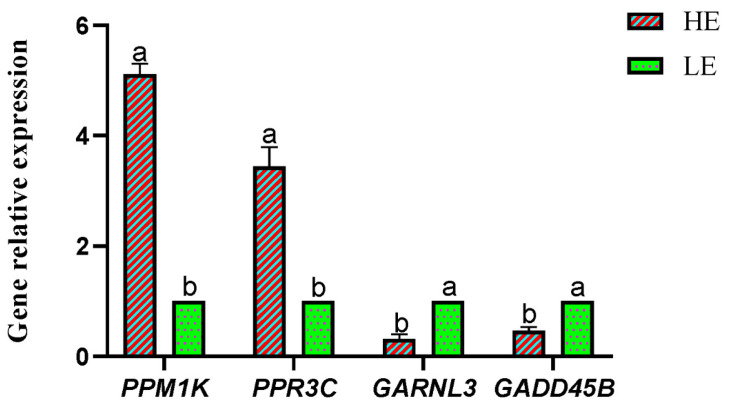
q-RT-PCR analysis of the significantly differentially expressed genes between high- and low-energy treatment Yushan pigs. HE = low-fiber content (high energy) treatment, LE = high-fiber content (low energy) treatment. Symbols of a,b, means a significant differential expression of the measured gene between LE and HE treatments.

**Figure 8 animals-14-02893-f008:**
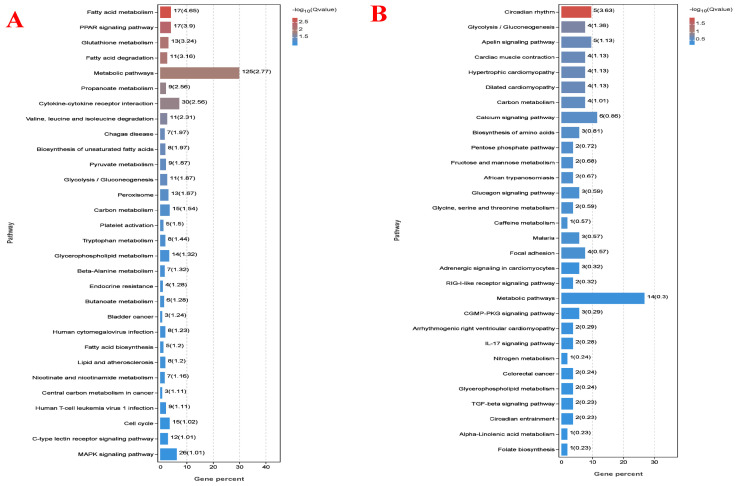
KEGG pathway analysis of differentially expressed genes between high- and low-energy treatment Yushan pigs. HE = low-fiber content (high energy) treatment, LE = high-fiber content (low energy) treatment. (**A**) Pathway analysis of up-regulated genes of low-energy treated livers compared with high-energy treated Yushan pigs. (**B**) Pathway analysis of down-regulated genes of low-energy treated livers compared with high-energy treated Yushan pigs.

**Table 1 animals-14-02893-t001:** Diet nutrient level and ingredients (dry matter basis).

Ingredients (%)	HE	LE
corn	65.76	48.66
soybean meal, 42% CP	13.75	8.25
wheat bran	10.4	18
rice bran	8	13
dicalcium phosphate	0.6	0.6
limestone	0.7	0.7
salt	0.4	0.4
mold inhibitor	0.1	0.1
antioxidants	0.04	0.04
corn bran	0	10
viatmin premix ^1^	0.15	0.15
trace mineral premix ^2^	0.1	0.1
Calculated composition		
ME, Mcal/Kg	3.11	2.87
CP, %	13.5	13.5
CF, %	3.42	4.36
NDF%	13.52	15.00
ADF%	5.90	6.34
NFE%	59.86	43.41
P, %	0.45	0.45
Ca, %	0.48	0.48

ME = metabolic energy, CP = crude protein, CF = crude fiber, NDF = neutral-detergent fiber, ADF = acid detergent fiber, NFE = nitrogen-free extracts, P = phosphorus, Ca = calcium. ^1^ Composition of vitamins per kilogram of final diet: vitamin A, 6614 IU; vitamin D3, 661 IU; vitamin E, 13.2 IU; riboflavin, 4.96 mg; vitamin B12, 0.02 mg; menadione, 2.4 mg; pantothenic acid, 16.9 mg; and niacin, 19.8 mg. ^2^ Composition of trace minerals in per kilogram of final diet: iron, 110 mg as iron sulfate; zinc, 110 mg as zinc oxide; manganese, 22 mg as manganese sulfate; copper, 11 mg as copper sulfate; iodine, 0.2 mg as potassium iodate; and selenium, 0.198 mg as sodium selenite.

**Table 2 animals-14-02893-t002:** Effects of high-fiber content diet treatment on productive performance of Yushan pigs *(n* = 9).

Items	HE	LE	SE	*p*-Value
Initial body weight (kg)	44.72	44.28	2.31	0.508
Final body weight (kg)	113.72	112.5	4.89	0.113
Average daily weight gain (ADG)/(kg)	0.60	0.59	0.07	0.673
Average daily feed intake (ADFI)/(kg)	1.90 ^b^	1.99 ^a^	0.03	0.036
Feed conversion ratio (FCR)	3.17	3.37	0.26	0.063

Letters ^a^ and ^b^ in each row describe significant differences between treatments at *p* < 0.05. HE = high-concentrate diet (high energy) treatment, LE = high-fiber content (low energy) treatment, SE = standard error.

**Table 3 animals-14-02893-t003:** Effects of high-fiber content diet treatment on carcass performance and meat quality of Yushan pigs (*n* = 9).

Items	HE	LE	SE	*p*-Value
Carcass weight/(kg)	84.42	79.81	1.33	0.026
Hind leg weight/(kg)	9.48	8.81	0.32	0.102
Diagonal length/(cm)	80.33	78.56	0.89	0.117
Dressing percentage (%)	74.28	73.55	1.38	0.627
Backfat thickness/(cm)	5.72	4.89	0.45	0.136
Lean percentage (%)	32.19	33.62	1.12	0.271
Fat percentage (%)	47.17	43.23	1.86	0.011
Skin percentage (%)	13.01	14.04	0.28	0.021
Bone weight percentage (%)	7.63	8.36	0.61	0.296
Eye muscle area/(cm^2^)	19.67	17.50	0.95	0.085
Shear force/(N)	48.57	48.36	10.48	0.985
Dropping loss (%)	2.95	2.60	0.49	0.327
Meat protein content (%)	16.34	16.88	1.22	0.684
Intramuscular fat (IMF) (%)	3.67	3.29	0.12	0.019
Inosine monophosphate (mg/L)	90.11	88.95	8.84	0.901

HE = high-concentrate diet (high energy) treatment, LE = high-fiber content (low energy) treatment, SE = standard error.

**Table 4 animals-14-02893-t004:** Effects of high-fiber content diet treatment on sarcous fatty acids composition of Yushan pigs (*n* = 9).

Items		HE	LE	SE	*p*-Value
Saturated	C14:0	1.38	1.27	0.04	0.058
	C16:0	25.6	24.5	0.34	0.032
	C17:0	0.25	0.22	0.01	0.140
	C18:0	15.61	10.87	4.76	0.377
	C21:0	0.27	0.29	0.03	0.532
	C20:0	0.44	0.43	0.03	0.83
	C23:0	0.43	0.42	0.07	0.851
Unsaturated	C16:1	2.47	2.33	0.16	0.446
	C17:1	0.14	0.12	0.01	0.101
	C18:1(cis)	38.97	40.74	0.58	0.038
	C18:1(trans)	3.57	3.11	0.18	0.048
	C18:2(cis)	9.08	9.59	0.63	0.46
	C20:1	1.05	1.03	0.08	0.808
	C18:3 (n3)	0.08	0.07	0.01	0.831
	C20:2	0.54	0.56	0.03	0.539

HE = high-concentrate diet (high energy) treatment, LE = high-fiber content (low energy) treatment, SE = standard error.

**Table 5 animals-14-02893-t005:** Differential analysis of plasma metabolites between high- and low-fiber content treatments of (*n* = 9).

Name		Rtmin	mz	FC	VIP	*p*-Value
down-regulated	N-(6-methoxypyridin-3-yl) thiophene-2-carboxamide	1.23	197.1	0.33	1.99	0.007
	SM (d21:0/13:0)	11.31	705.6	0.34	1.89	0.012
	DL-srginine	1.23	175.1	0.37	1.66	0.043
	Ciprostene	8.87	369.2	0.39	1.34	0.021
	Propionylcarnitine	2.65	218.1	0.4	1.34	0.006
	4,4′-dimethoxy[1,1′-biphenyl]-2-carbonitrile	4.77	240.1	0.49	1.29	0.003
	7-aminoflunitrazepam-d7	8.94	291.2	0.49	1.27	0.006
	1,2-dihydroxyheptadec-16-yn-4-yl acetate	8.44	349.2	0.49	1.25	0.008
	Desloratadine	8.94	311.1	0.49	1.09	0.039
up-regualted	Cyclic ADP-ribose	1.45	542.1	2	1.10	0.003
	7-methylguanine	1.86	166.1	2.01	1.24	0.011
	S-sdenosylhomocysteine	1.46	385.1	2.03	1.44	0.008
	(S)-10-hydroxycamptothecin	1.52	365.1	2.04	1.45	0.026
	N1-(2-cyanoethyl)-N1-cyclohexyl-4-bromobenzene-1-sulfonamide	3.45	371	2.06	1.46	0.014
	5-hydroxyindole-3-acetic acid	4.64	192.1	2.09	1.47	0.019
	L-Kynurenine	4.65	209.1	2.1	1.53	0.039
	R-1 methanandamide phosphate	5.12	442.3	2.11	1.58	0.019
	Diacerein	2.07	386.1	2.21	1.62	0.009
	Adenosine	2.66	268.1	2.29	1.67	0.011
	Carbamazepine-d10	2.43	247.2	2.36	1.72	0.035
	Triiodothyronine	1.22	651.8	2.03	1.76	0.021
	Hippuric acid	5.36	180.1	2.09	1.76	0.004
	PC (18:1/18:1)	5.53	786.6	2.18	1.89	0.044
	Leukotriene C4	6.7	626.3	2.26	1.90	0.009
	Diflucortolone pivalate	5.17	479.3	2.34	1.91	0.012

VIP = Variable importance in the projection. FC = fold change.

**Table 6 animals-14-02893-t006:** Effects of high-fiber content treatment on cecal bacterial α-diversity of Yushan pigs (*n* = 9).

Indexes	HE	LE	SE	*p*-Value
Sobs	1745	1655	41.67	0.147
Shannon	7.12	6.79	0.120	0.097
Simpson	0.97	0.96	0.001	0.128
Chao	1825	1740	43.76	0.193
ACE	1871	1783	45.00	0.188
Goods_coverage	0.99	0.99	0.001	0.236
Pielou	0.66	0.63	0.010	0.103
PD	244.4	236.2	5.620	0.299

HE = low-fiber content (high energy) treatment, LE = high-fiber content (low energy) treatment, SE= standard error.

**Table 7 animals-14-02893-t007:** Effects of high-fiber content treatment on cecal bacterial composition of Yushan pigs (level of phyla, *n* = 9).

Phylum	HE	LE	SE	*p*-Value
*Firmicutes*	53.63	50.31	3.07	0.296
*Bacteroidota*	18.64	23.95	1.54	0.031
*Spirochaetota*	7.72	11.16	1.07	0.028
*Actinobacteriota*	4.92	3.23	1.51	0.282
*Euryarchaeota*	3.05	3.28	1.04	0.827
*Patescibacteria*	3.08	2.34	1.58	0.643
*Proteobacteria*	3.33	2.07	0.30	0.035
*Verrucomicrobiota*	2.19	0.76	1.05	0.194
*Desulfobacterota*	0.57	0.41	0.11	0.151
*Planctomycetota*	0.39	0.29	0.22	0.677
Others	2.47	2.19	0.21	0.201

HE = low-fiber content (high energy) treatment, LE = high-fiber content (low energy) treatment, SE = standard error.

**Table 8 animals-14-02893-t008:** Effects of high-fiber content treatment on cecal bacterial composition of Yushan pigs (level of genera, *n* = 9).

Genera	HE	LE	SE	*p*-Value
*Bacteroides*	15.77	4.94	1.37	<0.001
*Lactobacillus*	2.50	11.29	2.24	0.001
*Rummeliibacillus*	6.03	3.85	2.21	0.339
*Prevotellaceae_UCG*	7.63	2.86	1.45	0.005
*Rikenellaceae_RC9*	4.36	3.19	0.71	0.121
*Faecalibacterium*	3.29	3.77	0.68	0.492
*Solibacillus*	2.15	4.18	1.89	0.300
*Methanobrevibacter*	2.39	3.94	0.97	0.129
*Enterococcus*	3.46	1.94	0.91	0.116
*Alistipes*	2.49	2.27	0.48	0.651
*Escherichia-Shigella*	2.72	1.85	0.39	0.043
*Corynebacterium*	1.42	2.70	1.22	0.311
*Ruminococcus*	0.86	2.85	0.78	0.021
*Clostridium*	2.66	0.95	0.52	0.005
*Akkermansia*	1.38	1.88	0.27	0.080
*Terrisporobacter*	0.52	2.16	1.02	0.126
*Romboutsia*	0.72	1.12	0.18	0.045
*Succinivibrio*	0.65	1.14	0.17	0.012
*Treponema*	0.72	0.73	0.14	0.918
*Phascolarctobacterium*	0.47	0.95	0.17	0.010
*Parabacteroides*	0.72	0.43	0.16	0.097
*Staphylococcus*	0.68	0.47	0.10	0.058
*Acinetobacter*	0.16	0.64	0.23	0.057
*Bifidobacterium*	0.24	0.52	0.08	0.005
*Fibrobacter*	0.15	0.51	0.31	0.252
*Flavobacterium*	0.21	0.32	0.09	0.189
*Butyricimonas*	0.03	0.02	0.01	0.162
others	35.46	37.48	2.96	0.503

HE = low-fiber content (high energy) treatment, LE = high-fiber content (low energy) treatment, SE = standard error.

## Data Availability

The data presented in the study were deposited in the NCBI Sequence Read Archive (SRA, http://www.ncbi.nlm.nih.gov/Traces/sra/, accessed on 1 June 2024), accession number PRJNA753017.

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
