# Peer review of "Role of Metabolomics and Metagenomics in the Replacement of the High-Concentrate Diet with a High-Fiber Diet for Growing Yushan Pigs"

_animals, 2024, doi:10.3390/ani14192893_

Round 1

Reviewer 1 Report

Comments and Suggestions for Authors

In this study, a large amount of bran was used to replace maize in feed formulations, which reduced fat deposition and improved liver metabolism in pigs. This important experimental result did not break through the long-term understanding of animal nutrition. In addition, the author's research does not seem to deeply reflect the changes and role of gut microbiota in fibre utilisation. Therefore, I don't think the innovation of this study is sufficient, although the manuscript provides some detailed experimental data through multi-omics analysis.

Other comments

1. The author did not provide the standard for the diet preparation. Why is there only one fixed feed formula in the long-term feeding experiment for pigs? Usually, feed formulas need to be adjusted according to the weight stage of pigs to meet their actual dynamic nutritional needs.

2. Since the LE treatment simultaneously reduced meat production, intramuscular fat content and back fat deposition in pigs, we cannot conclude that LE is more advantageous for pig production than HE. Identifying a suitable energy level for pigs is the key to further research.

3. The sample size for animal experiments is too small. Only nine replicates per group is not sufficient for statistical analysis of growth performance.

4. The QIIME (v1.7.0) is out of date for the microbiome analysis and the new version v2.0 should be used.

5. The clarity of the figures needs to be improved.

6. The current study lacks the necessary correlation analysis between different omics, e.g. a correlation analysis between microbiome and metabolome.

Author Response

In this study, a large amount of bran was used to replace maize in feed formulations, which reduced fat deposition and improved liver metabolism in pigs. This important experimental result did not break through the long-term understanding of animal nutrition. In addition, the author's research does not seem to deeply reflect the changes and role of gut microbiota in fibre utilisation. Therefore, I don't think the innovation of this study is sufficient, although the manuscript provides some detailed experimental data through multi-omics analysis.

Re. Thank you for the comment. Based on the comments, we corrected the introduction, material and methods description, added the correlation analysis in results, and corrected the discussion. We hope all these corrections could meet the requirement.

Other comments

  1. The author did not provide the standard for the diet preparation. Why is there only one fixed feed formula in the long-term feeding experiment for pigs? Usually, feed formulas need to be adjusted according to the weight stage of pigs to meet their actual dynamic nutritional needs.

Re. Thank you for the comment. Reasons for the diet preparation are mainly attributed into the following two reasons.

First, as a representative Chinese pig, Yushan pig has not been well studied which need more researches to determine the feed formula based on the weight stage and age. Therefore, in the present study, the only references we could take was the provincial standard 《DB36/T 171-2019—Yushan black pig》, in which the feed formula was not clearly mentioned based on the weight and age stages. Therefore, we just used the only diet in fattening period.

Second, differed from the growth cycle and speed of DLY, the growth period of Yushan black pig needed more than 12 months long to gain the sufficient weight. In our present study, growth period from 120-days old to 250-days old was a steady growth period for Yushan black pig, which needs no adjustment according to the weight stage. Therefore, we used only one diet preparation.

  1. Since the LE treatment simultaneously reduced meat production, intramuscular fat content and back fat deposition in pigs, we cannot conclude that LE is more advantageous for pig production than HE. Identifying a suitable energy level for pigs is the key to further research.

Re. Thank you for the comment, and sorry for giving an ambiguous conclusion between the LE and HE. For the purpose of this study and the domestic Chinese black pig rearing, reducing the back and abdominal fat content is of importance. In our present study, the back and abdominal fat content effectively reduced while lean percentage increased, although not significantly. Therefore, we corrected the conclusion that “The results indicated that increase diet fiber content could effectively reduce fat deposition process. Further researches to determine the optimum dietary energy level may provide a suitable energy level for Yushan pigs to gain a higher meat quality, lean percentage, and meat production.” We hope these corrections could meet the requirement.  

  1. The sample size for animal experiments is too small. Only nine replicates per group is not sufficient for statistical analysis of growth performance.

Re. Thank you for the comment. And sorry for the small experimental size. Two reasons make the sample size so small.

First, as a domestic Chinese pig, Yushan pig has a small total population and all of them are rearing in the conservation farm. For the purpose of declining the impact on breed conservation and regular production, combined with the consideration of the number of metabolomic, metagenomic, and transcriptomic analysis, total of 18 Yushan pig with the similar body weight were accurately selected for the experimental to reduce the standard error.

Secondly, in the previous study, we noted that : Thirty female pigs (42 d old) with an average initial body weight (BW) of 12.49 ± 1.45 kg were individually housed in raised pens at the Swine Nutrition Farm at Iowa State University (Ames, IA) for 19 d. Ten pigs were contained in each replicate. Therefore, we selected 9 pigs in each treatment.

Frame CA, Johnson E, Kilburn L, Huff-Lonergan E, Kerr BJ, Serao MR. Impact of dietary oxidized protein on oxidative status and performance in growing pigs. J Anim Sci. 2020 May 1;98(5):skaa097. doi: 10.1093/jas/skaa097

  1. The QIIME (v1.7.0) is out of date for the microbiome analysis and the new version v2.0 should be used.

Re. Thank you for the comment. QIIME 2 was now widely used in 16s and metagenomic sequencing analysis to acquire the accurate results. However, QIIME was also used in the recent years such as the following references.

Pal S, Vani G, Shivaji S, Donthineni PR, Basu S, Arunasri K. Characterising the tear bacterial microbiome in young adults. Exp Eye Res. 2022 Jun;219:109080.

Abdulhaq A, Halboub E, Homeida HE, Kumar Basode V, Ghzwani AH, Zain KA, Baraniya D, Chen T, Al-Hebshi NN. Tongue microbiome in children with autism spectrum disorder. J Oral Microbiol. 2021 Jun 22;13(1):1936434.

We further contact the sequencing company to gain the new results analyzed by QIIME 2, and only few results are differentially expressed and all the results are list in the supplementary file. We hope these corrections could meet the requirements.

  1. The clarity of the figures needs to be improved.

Re. Thank you for the comment. The order of figures was shown followed with metabolomics, metagenomics and genes expressions. Based on the comments, all figure captions are now corrected consistently, and expressed as “differential analysis between high and low energy treatment Yushan Pigs”. We added the correlation analysis between metabolomic and metagenomic results, hope all these corrections could meet the requirements.

  1. The current study lacks the necessary correlation analysis between different omics, e.g. a correlation analysis between microbiome and metabolome.

Re. Thank you for the comment, and sorry for not providing a correlation analysis. Based on the comments, we re-analyzed the data and made the correlation analysis between metabolomic results and metagenomic results.

“ Moreover, the interactive effects between significantly altered gut bacteria and the significantly altered metabolites were selected for correlations analysis, and all results are shown in Figure 6.

Microbial communities gathered into two main clusters. One was mainly consist-ed of Bacteroides, Rummeliibacillus, Enterococcus, Escherichia-Shigella, and Clostridium, which showed a significantly positive correlation with Ciprostene, and 7-Aminoflunitrazepam-d7, while a significantly negative correlation with 7-Methylguanine, S-Adenosylhomocysteine, Adenosine, and Hippuric acid. The other cluster was mainly consisted of Lactobacillus, Faecalibacterium, Ruminococcus, Bifidobacte-rium, and Fibrobacter, which showed a completely reverse correlation compared with the former one, and embodied in the significantly positive correlation with Hippuric acid, carbamazepine-d10, Diflucortolone pivalate, and Diacerein, while the significant-ly negative correlation with DL-Arginine, Desloratadine, and Propionylcarnitine. Spe-cifically, Bifidobacterium which was considered as the probiotic bacteria, showed a sig-nificantly positive correlation with all up-regulated metabolites in LE treatment com-pared with HE. No other significant correlations were observed.” Please check Line 378-393.

Reviewer 2 Report

Comments and Suggestions for Authors

The concept of this study is very interesting and has significant importance in the field of Animal Nutrition.

The objective ,methodology , major finings were presented well. However only the discussion part needs to improve with some own justification and I would like to suggest authors to carefully check each cited ref. In addition a final recommendation statement is important at the end portion of conclusion.  

Comments on the Quality of English Language

Fine

Author Response

The concept of this study is very interesting and has significant importance in the field of Animal Nutrition.

The objective, methodology, major finings were presented well. However only the discussion part needs to improve with some own justification and I would like to suggest authors to carefully check each cited ref. In addition a final recommendation statement is important at the end portion of conclusion.  

Re. Thank you for the comment. The discussion part has been re-written, we added the discussion on the correlation between omics, and the interactions between omics results and phenotypic traits. Please check the discussion part. We also add a recommendation statement at the end of conclusion. Please check the conclusion part.

Reviewer 3 Report

Comments and Suggestions for Authors

Title: Effects of Replacing High Concentrate Diet with High Fiber Content Brans on Productive Performances and Meat Quality of Growing Pig

The manuscript “Effects of Replacing High Concentrate Diet with High Fiber Content Brans on Productive Performances and Meat Quality of Growing Pig” suggested that higher dietary fibre content significantly reduced dietary energy provision and effectively decreased the backfat and abdominal fat content of Yushan pig through proliferating intestinal fibre-degradable bacteria, and up-regulating the hepatic lipolysis related gene expression. It is a well-written article with some interesting findings; however, there are some corrections before its acceptance for publication:

Line 17: What ADFI and FCR stands for?

Line 14-24: In simple summary, authors should describe their work simply and concisely to the public, therefore, it must be understandable to layman or farmer.

Line 25: The abstract must be a single paragraph of about 200 words maximum, these are around 272 words in the current manuscript. Secondly, there is no need to mention the headings, authors may follow them simply. Therefore, I invite authors to rewrite simple summary and abstract portions of the manuscript.

Line 60: In the introduction part, while describing the background of the topic, authors should discuss the effect of animal diet on meat composition, such as deposition of saturated or unsaturated (PUFA) fatty acids. I invite the authors to read and cite the following studies:

·       Tejeda, J. F., Hernández-Matamoros, A., Paniagua, M., & González, E. (2020). Effect of free-range and low-protein concentrated diets on growth performance, carcass traits, and meat composition of Iberian pig. Animals, 10(2), 273.

·       Lebret, B. (2008). Effects of feeding and rearing systems on growth, carcass composition and meat quality in pigs. Animal, 2(10), 1548-1558.

Line 109: Is it possible to place Table 1 in the supplementary file?

Line 100: How each pig was considered as a replicate, please explain in the materials and methods part.

Line 126: How about the mortality rate of the herd?

Line 128: Is it 16-h-long fasting or 12-h?

Line 145: Explain more about the determination of shear force values. Also, about cooking of the samples for taking WBSF readings.

Line 156: column oven?

Line 164: What is method 19?

Line 255: Please mention the superscript for post-hoc analysis test.

Line 267: Why sample size is so small? What about pooling of samples with large samples size?

Why the discussion section is so poor and weak. The authors need to explain how differential metabolites can influenced by different diets. The authors need to explain how fatty acids or metabolites are related with each other. Read and cite more references and make it strong.

The main drawback I find in the work is that all the metabolomic analysis has been performed with only nine samples per category: low dietary fiber content (high energy, HE) and high dietary fiber content (low energy, LE) treatments. While this might be a good preliminary study with such a small number of animals, the multivariate analysis of the data proposed (PLS-DA) is not robust and therefore these results should not be extrapolated, and this should be clearly indicated in the text.

There is 7% similarity with just one source, therefore, I suggest authors minimize it if possible to avoid plagiarism:

·       Liu, E., Xiao, W., Pu, Q., Xu, L., Wang, L., Mao, K., ... & Xue, F. (2022). Microbial and metabolomic insights into the bovine lipometabolic responses of rumen and mammary gland to zymolytic small peptide supplementation. Frontiers in Veterinary Science, 9, 875741.

Line 481: Authors should suggest some guidelines for future research, i.e., which aspect should be focused for future research.

Comments on the Quality of English Language

English grammar and sentence structure should be revised and corrected throughout the manuscript.

Author Response

The manuscript “Effects of Replacing High Concentrate Diet with High Fiber Content Brans on Productive Performances and Meat Quality of Growing Pig” suggested that higher dietary fibre content significantly reduced dietary energy provision and effectively decreased the backfat and abdominal fat content of Yushan pig through proliferating intestinal fibre-degradable bacteria, and up-regulating the hepatic lipolysis related gene expression. It is a well-written article with some interesting findings; however, there are some corrections before its acceptance for publication:

Line 17: What ADFI and FCR stands for?

Re. Thank you for the comment and sorry for not providing an ambiguous description. ADFI=average daily feed intake (ADFI), FCR= feed conversion ratio”. Because of no significant alteration was found for FCR, we deleted FCR in the simple summary. Please check Line 17.

Line 14-24: In simple summary, authors should describe their work simply and concisely to the public, therefore, it must be understandable to layman or farmer.

Re. Thank you for the comment. We deleted several specifical words in the simple summary, and we adjusted the expression. Simple summary has been corrected as “ The objective of this study was to investigate regulatory effects of replacing high concentrate diet with the high fiber content feed, which consisted of wheat bran, rice bran, and corn bran on the productive performances, meat quality, and fat acids composition. Results of our findings showed that, increasing dietary fiber content significantly increased average daily feed intake (ADFI), up-regulated carbohydrate-metabolism related metabolites, proliferated abundances of fiber-degradable microbial communities, such as Lactobacillus, and Bifidobacterium, and significantly up-regulated lipid metabolism, and cofactors and vitamins metabolism. Our findings indicated that higher dietary fiber content significantly reduced dietary energy provision, effectively decreased the backfat and abdominal fat content of Yushan pig through proliferating intestinal fiber-degradable bacteria, and up-regulating the hepatic lipolysis-related gene expression. This finding may provide an altered method for rearing domestic Chinese Pigs.”

Line 25: The abstract must be a single paragraph of about 200 words maximum, these are around 272 words in the current manuscript. Secondly, there is no need to mention the headings, authors may follow them simply. Therefore, I invite authors to rewrite simple summary and abstract portions of the manuscript.

Re. Thank you for the comment. Based on your suggestions, we corrected the abstract and the new abstract is list below.  “The objective of this study was to investigate regulatory effects of the high fiber content feed on the productive performances, meat quality, and fat acids composition. A total of 18 120-days-old Yushan Pigs with similar initial body weight was randomly allotted into high-concentrate diet (high energy, HE) and high fiber diet (low energy, LE) treatments for the determination of regulatory effects on productive performances, meat quality, and fatty acids content. Further, blood metabolomic, gut microbiota, and liver energy-related gene expression measurements were ap-plied to investigate the underlying mechanism. Results showed that the LE treatment significantly increased ADFI, while decreased carcass weight, fat percentage, and IMF. Metabolomic results showed high fiber treatment significantly down-regulated metabolites participated in lipid metabolism such as Cyclic ADP-ribose, and Hippuric acid, while up-regulated metabolites mainly enriched in nitrogen metabolism such as DL-Arginine, and Propionylcarnitine (P<0.05). Microbial results showed relative abundances of Lactobacillus and Bifidobacterium are significantly proliferated in high fiber feeding treatment (P<0.05). Transcriptomic results showed that genes mainly enriched into lipid metabolism are significantly up-regulated under the high-fiber dietary treatment(P<0.05). Conclusion: higher dietary fiber significantly reduced dietary energy provision, effectively decreased the backfat and abdominal fat content of Yushan pig through proliferating intestinal fiber-degradable bacteria, and up-regulating the hepatic lipolysis-related gene expression.” Please check abstract section.

Line 60: In the introduction part, while describing the background of the topic, authors should discuss the effect of animal diet on meat composition, such as deposition of saturated or unsaturated (PUFA) fatty acids. I invite the authors to read and cite the following studies:

  • Tejeda, J. F., Hernández-Matamoros, A., Paniagua, M., & González, E. (2020). Effect of free-range and low-protein concentrated diets on growth performance, carcass traits, and meat composition of Iberian pig. Animals, 10(2), 273.
  • Lebret, B. (2008). Effects of feeding and rearing systems on growth, carcass composition and meat quality in pigs. Animal, 2(10), 1548-1558.

Re. Thank you for the comment and suggestion. The description of the effect of animal diet on meat composition has been added. Please check Line 58-61.

Line 109: Is it possible to place Table 1 in the supplementary file?

Re. Thank you for the comment. Table 1 shows the mainly alterations of feed ingredients, so we considered it should list in the main text.

Line 100: How each pig was considered as a replicate, please explain in the materials and methods part.

Re. Thank you for the comment and sorry for not providing an accurate description. “Each treatment contains 9 pigs, each pig was considered as a replicate and reared in an individual pen with 220cm-long* 90cm-wide*130cm-height. Daily feed intake and daily weight gain were recorded individually. ” Please check Line 98-101.

Line 126: How about the mortality rate of the herd?

Re. Thank you for the comment. The mortality rate was 0 because we chose the 120-d-old pigs, and the immunity of domestic Chinese pig is outstanding. So that we did not mention the mortality rate here.

Line 128: Is it 16-h-long fasting or 12-h?

Re. Thank you for the comment. That is 12-h-long fasting treatment. Line 128.

Line 145: Explain more about the determination of shear force values. Also, about cooking of the samples for taking WBSF readings.

Re. Thank you for the comment. We corrected is as “ Specifically, the meat was first received cooking at the temperature of 72 °C, and cut into rectangular cooked meat sections (1 × 1 × 3 cm)when cooling into room temperature. Shear force was further measured perpendicular to the direction of fibers using a Texture Ana-lyser TA HD Plus (Stable Micro Systems Ltd, Surrey, UK) equipped with a Warner-Bratzler V-shaped shear blade (1.2 mm thick).” Please check Line 145-149.

Line 156: column oven?

Re. Thank you for the comment. Column oven was a thermostat column that heating the samples into gas status and then measured by absorption peak.

Line 164: What is method 19?

Re. Thank you for the comment and sorry for not providing an accurate description. 19 was a miss type number and has been removed. Line 165.

Line 255: Please mention the superscript for post-hoc analysis test.

Re. Thank you for the comment. The superscript for post-hoc analysis test and footnote” Letters a, b in each row describes significant differences between treatments at p < 0.05. ” have been added.

Line 267: Why sample size is so small? What about pooling of samples with large samples size?

Re. Thank you for the comment. And sorry for the small experimental size. Two reasons make the sample size so small.

First, as a domestic Chinese pig, Yushan pig has a small total population and all of them are rearing in the conservation farm. For the purpose of declining the impact on breed conservation and regular production, combined with the consideration of the number of metabolomic, metagenomic, and transcriptomic analysis, total of 18 Yushan pig with the similar body weight were accurately selected for the experimental to reduce the standard error.

Secondly, in the previous study, we noted that : Thirty female pigs (42 d old) with an average initial body weight (BW) of 12.49 ± 1.45 kg were individually housed in raised pens at the Swine Nutrition Farm at Iowa State University (Ames, IA) for 19 d. Ten pigs were contained in each replicate. Therefore, we selected 9 pigs in each treatment.

Frame CA, Johnson E, Kilburn L, Huff-Lonergan E, Kerr BJ, Serao MR. Impact of dietary oxidized protein on oxidative status and performance in growing pigs. J Anim Sci. 2020 May 1;98(5):skaa097. doi: 10.1093/jas/skaa097

Why the discussion section is so poor and weak. The authors need to explain how differential metabolites can influenced by different diets. The authors need to explain how fatty acids or metabolites are related with each other. Read and cite more references and make it strong.

The main drawback I find in the work is that all the metabolomic analysis has been performed with only nine samples per category: low dietary fiber content (high energy, HE) and high dietary fiber content (low energy, LE) treatments. While this might be a good preliminary study with such a small number of animals, the multivariate analysis of the data proposed (PLS-DA) is not robust and therefore these results should not be extrapolated, and this should be clearly indicated in the text.

There is 7% similarity with just one source, therefore, I suggest authors minimize it if possible to avoid plagiarism:

  • Liu, E., Xiao, W., Pu, Q., Xu, L., Wang, L., Mao, K., ... & Xue, F. (2022). Microbial and metabolomic insights into the bovine lipometabolic responses of rumen and mammary gland to zymolytic small peptide supplementation. Frontiers in Veterinary Science, 9, 875741.

Re. Thank you for the comment. And sorry for the un-sufficient discussion part. Based on the comment, we re-written the discussion, added the discussion of meat quality, and the mechanism of fat deposition. Please check the discussion part with high-lighten.

Line 481: Authors should suggest some guidelines for future research, i.e., which aspect should be focused for future research.

Re. Thank you for the comment. The suggestions have been added at the end of conclusion. Please check the conclusion part, high-lighten.

Round 2

Reviewer 1 Report

Comments and Suggestions for Authors

1. The author needs to articulate the limitations of this research in the manuscript. 

2. Provide VIP values for the differential metabolites. 

3. Can you provide a comparison of  the two groups regarding the functions of microbiota?

4. The explanation of the results needs to be more detailed.

Author Response

Dear reviewer, 

Thank you for the comment on the present article. Based on your suggestions, we corrected the article and responsed to the questions. Please check the additional file .

Best regards.

Reviewer 3 Report

Comments and Suggestions for Authors

The title of the article is too long and its length may be reduced for easiness of the readers. I would suggest "Role of Metabolomic and Metagenomic  in Replacing the High Concentrate Diet with High Fiber in Growing Yushan Pig".

Author Response

The title of the article is too long and its length may be reduced for easiness of the readers. I would suggest "Role of Metabolomic and Metagenomic  in Replacing the High Concentrate Diet with High Fiber in Growing Yushan Pig".

Response: The title has been corrected according to the suggestion. Please check the revised version. 

Best regards.